# Mosquitoes of Etiological Concern in Kenya and Possible Control Strategies

**DOI:** 10.3390/insects10060173

**Published:** 2019-06-16

**Authors:** Samuel Karungu, Evans Atoni, Joseph Ogalo, Caroline Mwaliko, Bernard Agwanda, Zhiming Yuan, Xiaomin Hu

**Affiliations:** 1Key Laboratory of Special Pathogens and Biosafety, Center for Emerging Infectious Diseases, Wuhan Institute of Virology, Chinese Academy of Sciences, Wuhan 430071, China; skarungu@yahoo.com (S.K.); atonet@live.com (E.A.); josephogallo@gmail.com (J.O.); carolinemwaliko@gmail.com (C.M.); 2University of Chinese Academy of Sciences, Beijing 100049, China; 3Mammalogy Section, National Museum of Kenya, P.O. Box 40658, Nairobi 00100, Kenya; benrisky@gmail.com

**Keywords:** *Aedes*, *Anopheles*, *Culex*, *Mansonia*, mosquito control, pathogens

## Abstract

Kenya is among the most affected tropical countries with pathogen transmitting Culicidae vectors. For decades, insect vectors have contributed to the emergence and distribution of viral and parasitic pathogens. Outbreaks and diseases have a great impact on a country’s economy, as resources that would otherwise be used for developmental projects are redirected to curb hospitalization cases and manage outbreaks. Infected invasive mosquito species have been shown to increasingly cross both local and global boarders due to the presence of increased environmental changes, trade, and tourism. In Kenya, there have been several mosquito-borne disease outbreaks such as the recent outbreaks along the coast of Kenya, involving chikungunya and dengue. This certainly calls for the implementation of strategies aimed at strengthening integrated vector management programs. In this review, we look at mosquitoes of public health concern in Kenya, while highlighting the pathogens they have been linked with over the years and across various regions. In addition, the major strategies that have previously been used in mosquito control and what more could be done to reduce or combat the menace caused by these hematophagous vectors are presented.

## 1. Introduction

The term “vector-borne” has become a commonly used term, especially in tropical and subtropical countries where emerging and re-emerging vector-related diseases frequently occur. One-sixth of human diseases is associated with vector-borne pathogens, with approximately more than half of the global population currently estimated to be in danger of contracting these diseases [1]. Hematophagous mosquitoes are the leading vectors among arthropods because of the significant role they play in disseminating microfilariae, arboviruses, and *Plasmodium* parasites that seem endemic to sub-Saharan Africa [2,3]. The majority of these pathogens are maintained in zoonotic cycles and humans are typically coincidental dead-end hosts with a none-to-minimal role in the cycle of the pathogen [4].

Mosquito vectoral ability is greatly influenced by the availability of conducive breeding grounds, which is in turn, influenced by the spatial heterogeneity as well as the temporal variability of the environment [5]. Mosquitos, pathogens, and hosts each endure and reproduce within certain ideal climatic conditions and changes in these conditions can greatly alter these pathogen transmission/competences. In this scope, temperature and level of precipitation are the most influential climatic components, but other factors such as sunshine length, sea level elevation, and wind have been shown to have considerable effects [6,7]. These vectors often adjust to changes in temperature by changing topographical distribution. For instance, the advent of malaria cases in the cooler regions of East African highlands may be attributed to climate change, which has led to an increase in mosquitoes in the highlands as they warm up [8]. Variability in precipitation may also have a direct influence on distribution of mosquito-borne diseases. When precipitation increases, the presence of disease vectors is also expected to rise due to the expansion of the existent larval habitat and emergence of new breeding zones [6]. Each mosquito species has unique environmental resilience limits dependent upon the availability of favorable aquatic larval habitats and the closeness of vertebrate hosts that serve as their source of blood meals. This reliance of mosquito species on aquatic environments is a constant part of their lifecycle and the availability of a suitable aquatic domain, which is a requirement for the development of eggs, larvae, and pupae, and basically determines the abundance of mosquito species.

Kenya represents a topographically diverse tropical/subtropical country which harbors a large diversity of mosquito species of public health importance. Many factors contribute to the extensive proliferation of mosquitoes ranging from global warming, sporadic floods, improper waste disposal, irrigation canals, presence of several lakes/rivers, and low altitudes around coastal regions. Consequently, an upsurge in emerging and re-emerging mosquito-borne pathogens over the years has been recorded with increased research efforts geared towards mosquitoes and their pathogens [9]. Increased urbanization, tourism, and international trade have led some of these species and pathogens to cross local and international borders to new territories. This dispersal poses both local and global health threats if proper mitigation measures are not put in place. Nevertheless, not all mosquito species are associated with human diseases, thus, this review highlights species of the main mosquito genera to which pathogens have been associated/detected and their countrywide distribution based on published data and reported cases. Additionally, this information provides a guide to proper mosquito control strategies by comparing the methods currently applied in the country and proposed alternative methods applied in other countries affected by mosquito disease burden.

## 2. Mosquito-Borne Disease Endemic Regions in Kenya

Kenya, as a tropical country, is among the most affected sub-Saharan regions with mosquito-related ailments. The countrywide distribution of mosquito species is, however, not well documented with most studies focusing on disease endemic regions. Some of these regions include Garissa, Mandera, and Turkana situated in the north and north-eastern parts of the country and characterized as arid and semi-arid regions (Figure 1A). During rainy seasons, flood water acts as breeding areas for mosquito species. Rift Valley fever outbreaks were recorded in this area in 1997/98 and 2006/07 [10,11]. Other viruses reported from the regions include West Nile virus (WNV), Ndumu virus (NDUV), Babanki virus (BBKV), and orthobunyaviruses isolated from the flood water *Aedes* and *Culex* species [4]. For decades, the coastal region of Kenya (Kwale, Kilifi, Mombasa, Lamu) has also reported multiple outbreaks resulting from mosquito-borne pathogens making it one of the most endemic regions. The contributing factors are majorly the low altitude which provides a conductive environment for mosquito breeding and high human population composed of both locals and global tourists. Mosquitoes of interest in this coastal region in terms of pathogen transmission include: *Aedes aegypti* that transmits dengue fever and chikungunya [12,13]; *Anopheles* species (*Anopheles gambiae*, *Anopheles arabiensis, Anopheles funestus*, *Anopheles merus*) which are associated with malaria and bancroftian filariasis sporozoites [14,15], and *Culex quinquefasciatus* [16] among others as demonstrated in Figure 1A–C.

Mosquito-related studies have been carried out in some of the mainlands surrounding lakes in Kenya, such as Lake Victoria and Lake Baringo due to the prevalence of malaria [17]. Other outbreaks such as Rift Valley fever near Lake Baringo also led to intensified surveillance on arbovirus mosquito vectors in the surrounding regions [4,18,19,20]. The favorable tropical climate and swampy areas around the lake regions throughout the year is among the factors favoring the high diversity of mosquito species [18,21]. A study by Lutomiah et al. [22] which recorded high numbers of *Culex pipiens* and *Culex univittatus* around the two lake regions additionally associated this abundance to highly humid conditions and the presence of migratory birds. Other lake regions in Kenya that are important for mosquito-related studies include areas around Lake Naivasha [23,24], Lake Bogoria, and Lake Nakuru [20,24] (Figure 1A,B).

Over the years, several malaria and arbovirus seropositive cases have been recorded in Kakamega and Busia located in the western part of Kenya [25,26,27]. In a study conducted in 2013, Kakamega and Busia showed high numbers of potential arbovirus vectors, with *A. aegypti* at 91.8% and 45.6%, respectively, of the total mosquitoes collected in the two regions [22]. Additionally, NDUV has been detected from *Culex* species collected from Busia [4]. Kakamega’s climate is characterized by high precipitation occurring throughout the year, most likely an influence of the Kakamega tropical rainforest while Busia is known to experience warm temperatures (annual mean temperature of 25 °C). These factors, among others, contribute to the high mosquito densities and consequent disease occurrence in the regions.

## 3. Overview of the Main Mosquito Genera of Public Health Importance in Kenya and Their Geographical Distribution

The main mosquito species associated with human etiologies in the country belong to four genera (*Aedes, Anopheles, Culex,* and *Mansonia*) as discussed below. Their distribution in Kenya has mainly been studied in disease endemic regions and as a result of outbreaks. These data are summarized in Table 1 and Table 2.

### 3.1. Aedes

*Aedes* represent a genus of mosquito that were originally known as tropical and subtropical mosquito species but have now been reported in almost all parts of the world apart from Antarctica. This genus comprises over 950 species normally identified by their black and white body coloration, preferential breeding in open water containers, and their daytime feeding, especially mornings and evenings [28]. Much interest in these mosquito species has been due to their role as major vectors of arboviruses across the globe. For instance, in various islands of the south pacific, *A. polynesiensis*—a primary vector of *Wuchereria bancrofti*—has also been identified as an endemic dengue vector [29]. *Aedes scutellaris*, a native species of Papua New Guinea, has also been linked to dengue outbreaks around the region and other pacific islands [30,31]. *Aedes aegypti* and *A. albopictus* of African and Asian origins, respectively, are globally known for their invasive nature and role in transmitting viruses that cause chikungunya, dengue fever, yellow fever, Zika, and encephalitis [32]. The success of these species in virus transmission has been attributed to three mechanisms: interaction of vector and the host, transovarial transmission, and sexual mating [28,33].

Kenya harbors more than 80 different *Aedes* species (http://www.wrbu.org) but not all have been associated with the spread of mosquito viruses. An important species is *A. ochraceus* which is highly distributed in the arid regions of north-eastern Kenya and linked to the spread of Rift Valley fever virus (RVFV), NDUV, Bunyamwera virus (BUNV), Babanki virus (BBKV), Sindbis virus (SNBV), and Semliki Forest virus (SMFV) in the region [4,11,34,35,36,37,38]. Additionally, this species has also been reported to be associated with RVFV, dengue virus (DENV), and chikungunya virus (CHKV) in coastal counties (Kwale, Kilifi, Mombasa, Lamu) [11,33,39], and some parts of in the western region [22]. *Aedes mcintoshi* and *A. sudanensis* are also important arbovirus vectors occasionally reported in coastal and north-eastern regions [4,11,14,22,33,36,40], while *A. circumluteolus* have been reported in Busia and Garissa counties [4,22,36]. Other *Aedes* secondary vectors of arboviruses such as: *A. tricholabis* [4,33], *A. Albicosta* [33], *A. fulgens* [33], *A. fryeri* [33], *A. pembaensis* [33], and *A. Luridus* [4] have been identified in various parts of the country and mostly along the Kenyan coastline. *Aedes africanus* and *Ae. keniensis* sampled from Baringo county represent the only field collected *Aedes* spp. in which successful yellow fever virus (YFV) has been detected in Kenya [41]. Moreover, other *Aedes* spp. such as *A*. *vittatus* and *A*. *bromeliae* have shown competence to DENV and YFV, even though no field detection has been recorded [42,43].

*Aedes aegypti*, the principal vector of dengue, chikungunya, and other emerging arboviruses is also widely distributed in Kenya [4,33,44] but not uniformly, with more occurrence being recorded in the lowlands [22]. It exists in two forms that were found coexisting sympatrically in Rabai, along the coast of Kenya [45]: the domestic, light-colored *A. aegypti aegypti* and the sylvatic form, *A. aegypti formosus,* which is dark in color [45,46]. In addition to morphological differences, definite behavioral variabilities have also been observed between the two forms of *A. aegypti* [46]. They include a longer developmental time and indoor breeding tendency of the domestic form as opposed to the ancestral sylvatic form that usually breed in forests tree holes and develop within a shorter period [47,48]. This explains the occurrence of *A. aegypti formosus* in vegetated ecosystems and previous documentation in western Kenya near Kisumu city and in the Kakamega forest [46]. Additionally, the sylvatic type is predominantly zoophilic as compared to the domestic type, which prefers humans for their blood meals.

Rapid urbanization in Kenya, facilitated by the increase in both domestic and international trade has led to major ecological and social changes [14]. These transformations have attracted *A. aegypti* to urban centers with up to three times the population size compared to rural areas due to the accumulation of non-biodegradable human-made containers that serve as conducive breeding sites to these vectors [12,49,50]. It is, thus, evident that if effective vector control strategies are not enhanced, the rate of arbovirus disease transmission is bound to increase, especially in urban centers.

Sporadic arboviral disease outbreaks associated with *Aedes* spp. have occurred over several decades in Kenya of which some have shown a tendency to recur in the same or new populations. For example, dengue fever in 1982 (Malindi, coastal region) [51], 2011 (Mandera, north-eastern) [52], and 2013–2014, 2017 (coastal region counties) [12,53]; Rift Valley fever in 1997–1998 (Garissa, north-eastern) [10], 2006–2007 (Garissa and Baringo) [11], 2014–2015, 2018 (north-eastern) [54]; yellow fever in 1992–1993 (Kerio Valley, Rift Valley region) [41]; chikungunya in 2004 (Lamu, coastal region) [55], 2016 (Mandera, north-eastern) [56], and 2018 (Mombasa, coastal region) [13].

### 3.2. Anopheles

This genera comprises mosquito species that are commonly identified by long palps, dark-brown body coloration, discrete black and white scales located at the wings and a characteristic resting position (stomach area pointing upwards) [63]. Over 460 anopheline species have been recognized across the world, with approximately 7% known to transmit *Plasmodium* parasites that cause malaria in humans [64]. Malaria is endemic in 91 countries, with about 40% of the world’s population at risk. The latest World Health Organization (WHO) data showed that in 2017, there were 219 million cases of malaria and estimated 435,000 malaria deaths across the world. Out of these, Africa accounted for the most cases and deaths (92% and 93%), respectively [65].

Africa is home to more than 140 anopheline species, of which six of them are primary vectors of malaria parasites (*P. falciparum*, *P. vivax*, *P. ovale,* and *P. malariae*) [66,67]. They include *Anopheles gambiae*, *Anopheles arabiensis*, *Anopheles funestus*, *Anopheles nili*, *Anopheles moucheti,* and *Anopheles coluzzii* [67,68]. However, other than the primary vectors that contribute to 95% of malaria transmission in Africa, secondary *Anopheles*’ vectors have also been recorded and contribute 5% of the overall transmission. For instance, *An. pharoensis* and *An. ziemanni* were found to harbor the malaria parasite *P. vivax* in Ethiopia at irrigated rice plantations [69]. In Tanzania, *An. ziemanni*, *An. Coustani,* and *An. squamosus* have also been linked to malaria transmission as secondary vectors [70].

In Kenya, *Anopheles* spp. are a major concern in the public health sector due to their role as malaria transmission agents. In a country-wide study by Okiro et al. [71], out of 166,632 hospital admissions, western Kenya had the highest number of cases (70%) with the Rift Valley highlands (45%) and Kenyan coast (22%) coming in at 2nd and 3rd, respectively. Each year, approximately 3.5 million new malaria cases and 10,700 deaths are recorded, with those residing in the western part of Kenya being at high risk of contracting the disease [72].

The main vectors of malaria in Kenya include the *An. gambiae* complex (*An. gambiae* s.s, *An. arabiensis, An. merus*) and *An. funestus*. These transmit all four *Plasmodium* parasites with *P. falciparum* being the most common, accounting for 99% of all malaria cases within the country [73]. *Anopheles arabiensis* and *An. gambiae* have been shown to coexist sympatrically, especially in the coastal counties Kilifi and Kwale, and Nyanza counties Kisumu and Siaya [74]. However, evidence has shown that over time *An. arabiensis* started becoming the more predominant vector where the two coincide [75,76,77]. *Anopheles merus* is largely distributed within a 25 km radius from the Kenyan coast due to its need to breed in saline water [71,78]. See Figure 1C and Table 2.

Secondary vectors of malaria are also wide spread within the country. For instance, *An. pharoensis* sampled from a rice irrigation scheme in Mwea, central Kenya, constituted 1.3% and 0.68% *P. falciparum* by ELISA (enzyme-linked immunosorbent assay) and dissection methods, respectively [79]. This species has also been reported in western region highlands [80]. *Anopheles nili* has been identified in 26 scattered sites along the coastal areas: Trans Nzoia, Taita and Taveta area, Kaimosi Forest in Vihiga county, Mwea tebere rice plantation, and Thika [81]. *Anopheles coustani* and *An. ziemanni* sampled from Taveta and western regions, respectively, have also been implicated as potential secondary vectors [14,82]. Regardless of the country-wide distribution of these potential secondary vectors, their actual role in the transmission of malaria parasites in the country is not well known.

Kenya has been stratified into four malaria epidemiological zones with risk factors determined usually by temperature, altitude, rainfall patterns, and malaria prevalence [83]. These zones are: a) The endemic areas in coastal regions and Lake Victoria in the western region. These areas are characterized by altitudes ranging from 0–1300 m above sea level and stable high malaria prevalence throughout the year. b) Highland epidemic prone areas that experience seasonal malaria transmission with considerable year-to-year variations around the western highlands and between 5–20% prevalence. c) Seasonal malaria transmission areas which include arid and semi-arid areas in the northern and south-eastern parts of the country where they experience short episodes of intense malaria transmission during the rainy seasons. d) Low malaria risk areas in the central highlands of Kenya such as Nairobi, where temperatures are too low for the survival of malaria *Plasmodium* parasites [83]. However, with changes in annual climatic patterns and increase in global warming, previously malaria-free regions may soon become endemic areas. For instance, mosquito species such as *An. arabiensis* which were not known to inhabit central Kenya highlands are now being reported [66].

Although the government of Kenya has established various measures that have seen a reduction in malaria prevalence especially in major endemic regions [83], this has not been the case for some counties. In October 2017 and February 2018, malaria outbreaks hit the arid counties of Marsabit, Baringo, and West Pokot leading to numerous hospitalization cases and deaths [84]. This calls for more cohesive policies in the fight against malaria and their respective anopheline vectors.

Apart from the spread of malaria, *Anopheles* spp. are associated with spread of other vector-borne diseases such as: canine heartworm (*Dirofilaria immitis*) common in the Americas [85,86], lymphatic filariasis, and O’nyong’nyong fever. The first case of O’nyong’nyong virus (ONNV) was reported in Uganda in 1959 and spread through the Victoria basin to western Kenya by 1960 [87]. From this time onwards, this virus has repeatedly been isolated from *An. funestus* and *An. gambiae* in different localities within Kenya providing strong evidence that these two mosquito species are indeed the primary vectors [59,88]. Other mosquito-borne viruses detected from *Anopheles* spp. in the country include: Ngari virus (NRIV) in the Tana River (*An. funestus*) [4] and BUNV in Homabay (*An. gambiae*) [20], as summarized in Table 1. Additionally, *An. funestus* and *An. gambiae* are also associated with the spread of lymphatic filariasis in the Kenyan coast from Lamu to Kwale where ecological factors favor its transmission [89,90,91,92].

### 3.3. Culex

Drab in color, the *Culex* genera comprises up to 1000 species with a fairly weak flying tendency and generally found in all zoogeographical areas ranging from the tropics to cooler temperate regions [96]. Due to their proximity to human settlements, these species are often referred to as common house mosquitoes. They have also been shown to prefer their blood meals from birds that feed at dawn and dusk [97]. Among the multiple *Culex* spp., *Cx. pipiens* complex is the most widely distributed across the globe. It comprises *Cx. Pipiens* L, *Cx. quinquefasciatus* Say, *Cx. pipiens pallens* Coquillett, *Cx. australicus* Dobrotworsky and Drummond, and an autogenous *Cx. pipiens* subspecies *Cx. pipiens molestus* Forskal. Worth noting is that the complex species are not easily distinguished morphologically and can only be separated through a combination of molecular techniques, and behavioral and physiological traits [98,99].

*Culex* spp. are globally known as vectors of human, avian, and animal pathogens such as lymphatic filariasis primarily vectored by *Cx. pipiens* [100,101]. This disease is caused by a nematode called *W. bancrofti* which is currently in more than 52 countries where 856 million people live in disease endemic regions (South-East Asia (66%) and African regions (33%)) being at high risk [102]. *Culex* spp. are also associated with transmission of viruses, with WNV being the most common. The virus is often amplified by *Cx. pipiens quinquefasciatus* and maintained in a cycle involving humans and birds although other species of the *pipiens* complex have shown competence in spreading the virus [60,97,103,104]. Originally isolated in Uganda in 1937, this virus has spread globally and has been reported in almost all continents [105]. Other pathogens vectored by *Culex* spp. include the Saint Louis encephalitis virus (SLEV) [106], equine encephalitis virus (EEV) [107], dog heartworm (*Dirofilaria immitis*), SNBV, *Plasmodium relictum* (bird malarias) [103], and RVFV. Whether or not Zika virus is vectored by *Culex* spp. has been a topic of debate. Sammy et al. [108] noted the possibility of *Cx. quinquefasciatus* playing a vectorial role in Zika prone areas in Brazil due to their higher abundance compared to *A. aegypti,* which is a commonly known Zika transmission vector. Later on, Amraoui et al. [109] and Huang et al. [110] rejected this possibility. However, a recent study by Phumee at al. [111] evidenced that vertical transmission of Zika virus by *Cx. quinquefasciatus* was indeed possible.

According to the Walter–Reed Bioinformatics Unit (WRBU), Kenya harbors approximately 42 *Culex* spp. (http://www.wrbu.org). The ones with the most public health significance include members of the *Cx. pipiens* complex, *Cx. pipiens* L. and *Cx. quinquefasciatus Say,* whose success in colonizing both urban and rural environs have been attributed to their larval ability to develop in a variety of aquatic habitats [14]. These mosquitoes are highly distributed in Kenyan coast counties and associated with the spread of bancroftian filariasis [90,91,95] and cases of RVF [11,112]. A study by Atoni et al. [17] recorded high numbers of *Cx. quinquefasciatus* in Kwale county, which had abundant vertebrate viruses based on metagenomic analysis. These species have also been implicated in entomologic studies focusing on WNV, SNBV, and NDUV in Garissa county [38,39,60], RVF in Baringo, Garissa, and Nakuru counties [11,19,60], and Usutu virus in Kisumu [4].

*Culex univittatus* is another common *Culex* spp. distributed within the country. Its first vertical natural transmission of WNV was evidenced in the Rift Valley province in a study by Lanciotti et al. [61]. In Baringo county, it has been implicated as a vector of the RVFV [19] and BUNV in the Lake Victoria region [20]. In an entomologic survey by Lutomiah et al. [22], on the abundance of potential mosquito vectors, this species was found to be highly abundant in Kisumu (37%) and Naivasha (21.6%), even though no pathogen screening was conducted.

Other disease important *Culex* spp. in the country include *Cx. bitaeniorhynchus, Cx. cinereus, Cx. poicilipes*, *Cx. rubinotus*, *Cx. Vansomereni,* and *Cx. zombaensis* in which various etiological pathogens have been isolated as summarized in Table 1. Vector competence involving some of these species has also been witnessed. For instance, in a study by Turell et al. [23], *Cx. zombaensis* collected around the Lake Naivasha region showed high competence for RVFV with an 89% infection rate.

### 3.4. Mansonia

The *Mansonia* mosquito species belong to the sub-family Culicinae and are normally characterized by their large body size and asymmetrical wing scale structure with sparkling on their wing veins and legs. They show some resemblance with *Aedes, Culex,* and *Coquillettidia* mosquito but the simplicity of their tarsal claws in structure and a truncated abdomen in females sets it apart [113]. Furthermore, *Mansonia* preferably breed in ponds and permanent waters with aquatic plants where larvae can burrow into the dead plants at the bottom or cling to the roots of the live ones [114]. Its genera comprise 25 globally distributed species in two subgenera: *Mansonioides* (10 species) and *Mansonia* (15 species).

The distribution of these species comes with increased vectorial role of human etiologies. *Mansonia titillans,* a common species in the Americas, is a known vector of Venezuelan equine encephalitis [115]. Recently, Argentina detected SLEV and BUNV for the first time from this species, proving its competent nature to transmit the viruses [116]. In south-east Asia, *Mn. bonneae* and *Mn. dives* have been linked to various cases of Brugian and lymphatic filariasis [117]. Africa, as one of the most affected regions of vector-borne diseases, harbors two main and probably the most common *Mansonia* spp. (*Mn. africana* and *Mn. uniformis*). These species have been reported to vector *W. bancrofti*’s lymphatic filariasis [118,119], RVF [34], Zika [120], among other diseases. Kenya is not an exception, as studies have shown that the two *Mansonia* spp. are distributed within the country and harbor pathogens of human health importance [4,11,36,40,60].

LaBeaud et al. [60] showed that *Mansonia* mosquitoes could have significantly contributed to the vectorial transmission of RVF in the 2006–2007 outbreak in the north-eastern region. Out of the 12,080 mosquitos collected during the study, 682 were identified to be *Mansonia* spp. Further investigation based on molecular screening identified three out of eight pools of *Mansonia* spp. to be positive for RVFV. *Mansonia,* like many other female mosquitoes, feed on vertebrate blood for egg development. These vertebrates can lead to amplification of a disease in case of outbreaks. Virus screening from blood-fed *Mansonia* spp. by Lutomiah et al. [40] during this same RVF outbreak, showed that goats and sheep were the greatest amplifiers of the virus.

Owing to the fact that *Mn. uniformis* and *Mn. africana* preferably breed around permanent water bodies, flooded lagoons, and swamps, they have been shown to dominate lake regions in the country. A study by Lutomiah et al. [22] on the abundance of mosquito vectors in various ecological zones in the country revealed high abundance of *Mansonia* spp. in Baringo (71%) and Kisumu (23%), the home places of Lake Baringo and Lake Victoria. Interestingly, it is here (Baringo) that positive strains of RVFV and NDUV had earlier been detected in *Mn. uniformis* and *Mn. africana* [11,36]. This was further evidenced by Ajamma et al. [18], where these species dominated in abundance in the islands and mainlands of both Lake Victoria and Lake Baringo.

Even though Zika virus and *W. bancrofti* have been detected from *Mansonia* spp. in some African states [118,120], this has not been the case in Kenya. In a study at the Kenyan Tana delta on vectorial potential of *Mansonia* spp. to transmit *W. bancrofti;* all 236 species collected tested negative on PCR (polymerase chain reaction) analysis [121]. However, the study recommended further assessment to be done by infecting *Mansonia* spp. experimentally with *W. bancrofti* microfilaria to identify whether or not they can support development of microfilaria up to the infective stages.

## 4. Mosquito Control Strategies in Kenya

Regardless of the fact that various control measures have for long been implemented both locally and internationally, mosquitoes continue to be a public health nuisance/concern in Kenya as well as many other African countries. This raises the questions what control measures have been undertaken in the country and what more needs to be done. Table 3 summarizes the common mosquito control strategies in Kenya and proposed alternatives in a bid to strengthen the integrated vector management (IVM) programs.

### 4.1. What Has Been Done

#### 4.1.1. Entomologic Surveillance

A successful vector control program requires pre-entomologic investigations which inform on the abundance of mosquito vectors in different regions and proper control strategies. The Kenyan government through the president’s malaria initiative (PMI) has in place a surveillance program mainly focusing on anopheline mosquito species, the main vectors of malaria parasites [83]. Between 2008 and 2015, the program surveyed over 16 sites in the western part of Kenya, in which three main malaria vectors (*An. arabiensis, An. gambiae* s.s., and *An. funestus*) were found to dominate. Data have shown fluctuations in these species over time in terms of abundance, for example, in 2008 *An. arabiensis* was the most abundant species in the region, but in 2016, *An. funestus* s.s. was seen to dominate the overall species composition by 93% based on genetic sequencing results [83].

Other governmental and non-governmental organizations in collaboration with the international community have played various roles in conducting entomological investigations involving vectors of arboviruses. The Kenya Medical Research Institute (KEMRI) in collaboration with the United States Medical Research Unit (USAMRU) recently conducted a mosquito investigation in Mandera, north-eastern Kenya focusing on chikungunya vectors where *A. aegypti* was found in abundance mostly in underground tanks and outdoor open water containers [58]. Other studies include entomological surveys involving dengue vectors in Mombasa [12], RVF vectors in Garissa [11], and Baringo [19] among others.

Nevertheless, there is still need for country-wide surveillance of mosquitoes covering both disease endemic and non-endemic regions in the country.

#### 4.1.2. Use of Synthetic Pesticides

Synthetic pesticides such as pyrethroids, carbamates, organophosphates, and organochlorines have been applied as adulticides and larvicides of mosquitoes in the country for decades [122]. Despite their fast mode of action against mosquito vectors, most have been shown to have adverse effect on the environment with adverse consequences to non-target organisms. As a result, the majority have been banned from use by the Ministry of Health. For instance, DDT (dichloro-diphenyl-trichloroethane) an organochlorine pesticide, was banned in 1986 due to the high association with hormone disruption and linkage to psychological problems in humans. Currently, pyrethroids are the only allowed pesticides for use in indoor residual spraying (IRS) and long-lasting insecticide nets (LLIN) in Kenya [123]. Use of this intervention has yielded significant success in the control of mosquito vectors over a considerable period of time, with up to 68% malaria mortality reduction in Africa [124]. However rampant application of pyrethroids in mosquito control interventions has resulted in development of resistance among the vectors which risks retrogression on the achieved success [122]. The highest records of resistance have been reported in the western [125], coastal [126], and central parts of Kenya involving *An. gambiae*, *An. Funestus,* and *An. arabiensis* [122].

#### 4.1.3. Insect Growth Regulators (IGRs)

Insect growth regulators (IGRs) are potent insecticides formulated from chemical or botanical sources. Unlike other synthetic pesticides, IGRs are considerably safer for non-target organisms including humans, as they interfere with specific biochemical pathways required for insect growth and development [127,128]. Methoprene and pyriproxyfen represent some globally applied IGRs and are characterized as juvenile hormone analogs. Generally, these compounds mimic the mosquito juvenile hormones by binding to the receptors of the immature mosquito stages preventing a successful metamorphosis [128].

In Kenya, various tests involving IGRs have been carried out. For instance, impregnating bed nets with pyriproxyfen in western Kenya showed a considerable reduction of pyrethroid-resistant *An. gambiae* s.s. after three days post-collection [129]. Moreover, application of *Camellia sinensis* leaf extracts in a lab-based experiment inhibited larval development and adult emergence of *An. Arabiensis* and *An. gambiae* s.s. [130]. These prove that integration of IGRs in large-scale mosquito control could boost the IVM programs in the country. However, like many chemical insecticides, development of resistance is of much concern among the IGRs. Case scenarios include: *Ochlerotatus nigromaculis* (Ludlow) resistance to methropene [131] and *Cx. quinquefasciatus* to pyriproxyfen in California, USA [132]. Nevertheless, the reported resistance among the IGRs is considerably low compared to other chemical insecticides.

### 4.2. What More Needs to Be Done?

In the wake of IVM, it is clear that more control measures need to be established if Kenya and Africa at large are to win the war against these hematophagous vectors.

#### 4.2.1. Public Awareness on Environmental Management

There is a need for public awareness and involvement in the venture to control mosquito vectors. This can be achieved by establishing public–private stakeholder partnerships with a mandate to educate and empower both rural and urban populations on effective mosquito control measures. Among the areas of emphasis would be the importance of environmental management and general hygiene for sustainable mosquito control. A breakdown of the requisite empowerment content includes: (a) breaking the mosquito lifecycle chain by clearing all breeding sites such as overhead irrigation ditches and use of tight lids for water drums, jars, and overhead tanks; (b) proper solid waste management such as collecting discarded car tires, tins, and bottles for recycling; and (c) proper house designs to reduce entry of mosquitoes, for instance the use of wire meshed windows.

The strategy of engaging community participation in vector management has previously been successful in Malindi and Siaya counties during a six-year program (2006–2011) and can, therefore, be replicated in other parts of Kenya [133].

#### 4.2.2. Embracing Biological Controls

As a consequence of environmental degradation and acquired mosquito resistance associated with synthetic pesticides, bio-controls should be embraced as an alternative control strategy. Bio-pesticides, unlike chemical alternatives, provide a safer mode of action with regard to environmental protection. They include: natural predators, entomopathogens, and *Wolbachia* endosymbiotic bacteria.

##### Use of Natural Predators

Culicinae larvae and pupa stages serve as food to many aquatic organisms including larvivorous fish, omnivorous copepods, amphibians, and some water bugs, thus, reducing their numbers wherever they co-exist. Introduction of such organisms in various mosquito-breeding sites has been demonstrated to significantly reduce the vector populations [134]. For instance, introduction of copepods in northern Vietnam in 1993 in a bid to control dengue vectors *A. aegypti* is the greatest bio-control success involving copepods ever recorded [135]. In western Kenya, use of larvivorous fish *Oreochromis niloticus* L. was shown to reduce *An. gambiae* s.l. and *An. funestus* by 94% in abandoned ponds after 15 days. These, among other cases, show the potential for use of mosquito predators as part of the integrated vector management in the country [136]. Nevertheless, they are not without limitations, as they can only be used in certain limited aquatic habitats. With reference to amphibians and fish, their applicability in control of *Aedes* spp. in some urban breeding environments such as waste tires and poorly discarded containers, is not suitable [134].

##### Large-Scale Applications of Entomopathogens

Application of entomopathogens to control mosquito vectors has recorded tremendous success where it has been applied. *Bacillus thuringiensis israelensis* (*B.t.i*) and *Lysinibacillus sphaericus (L.s)* bacterial strains have been applied and are available as commercial products to control the mosquito larvae. The toxicity of these bio larvicides is attributed to their toxin genes (*B.t.i*—crystal toxins, *L.s*— binary toxins) especially during sporulation [137]. *B.t.i* toxin has a broad spectrum of activity against *Aedes*, *Anopheles,* and *Culex* spp. as compared to *L.s,* which tends to be more specific against *Culex* and to some extent *Anopheles* spp. [138,139]. Various trials have been conducted in Kenya involving imported commercial *B.t.i/L.s* products such as Culinex^®^ Combi Tab plus (Mannheim, Germany) in Malindi [140] and FourStar^®^ briquets (California, USA) in the western Kenyan highlands [141] with successful larvicidal activity. Nevertheless, the prices of these imported products have been a limiting factor especially for large-scale applications. In recent work by Karungu et al. [142], larvicidal strains of *B. thuringiensis* and *L. sphaericus* were successfully isolated from Kenyan soil samples. These local strains can be used to produce a local Kenyan based bio-larvicide using cheap home-based agricultural products as raw materials for solid-state fermentation (SSF) and submerged fermentation (SmF).

Entomopathogenic fungi, *Beauveria bassiana* and *Metarhizium anisopliae,* are other natural mosquito control pathogens and have been extensively applied to control multiple insect pests. Their mode of action involves germination of the fungal conidia on the mosquito surface, which further penetrate the insect cuticle leading to its death [143]. Successful field applications have been demonstrated in various countries to control *A. aegypti* [144], *A. albopictus* [145] and malaria mosquito vectors [146]. A couple of these fungi such as *Metarhizium anisopliae* ICIPE 7 and *Metarhizium anisopliae* ICIPE 20 have been isolated from Kenya and tested in the control of various insect pests [147,148]. Their success, especially in tick control programs, has seen their commercialization and official product registration. Application of such technology in the large-scale control of mosquitoes in Kenya would result in a great boost to the IVM initiatives.

##### Incompatible Insect Technique (IIT)

This technique involves the use of *Wolbachia* endosymbiotic bacteria known to infect at least 60% of all insect species [149]. Its success in mosquito control is dependent on how many males are infected with *Wolbachia* in that mating between an infected male and an uninfected female forms inviable eggs, a condition called cytoplasmic incompatibility (CI) [150]. Field trials involving IIT have been conducted in various countries globally such as Australia involving *Ae. aegypti* infected with *wmel Wolbachia* strain [151], USA involving *Ae. albopictus* [152] and French Polynesia [153] involving *Ae. polynesiensis* with promising results. These marked successes are a challenge to Kenya and many sub-Saharan countries where this technique is poorly/not established. As part of IVM programs, the release of *Wolbachia* infected males into areas with high mosquito incidences should be part of the initiatives.

#### 4.2.3. Genetically Engineered Mosquitoes

Some of the techniques involved here include: the sterile insect technique and release of insects carrying dominant lethality.

##### The Sterile Insect Technique (SIT)

This strategy involves genetic suppression of the male mosquito through irradiation, which causes chromosomal aberrations in the germ cells. As a result, no progeny is produced after the sterile male mates with a wild female mosquito [154]. A few projects involving the release of sterile male mosquitoes have been conducted primarily targeting *A. aegypti* and *A. albopictus*, but have encountered some challenges. First, construction of the infrastructure for mass rearing of mosquitoes is not a simple/cheap endeavor. Secondly, there has been poor performance of the sterile males in reducing the vector populations in target regions [134]. As a result, other mechanisms such as contamination and auto-dissemination of a growth regulator (pyriproxygen) in breeding zones to prevent emergence of adults from the laid eggs should be applied concurrently with SIT to improve the performance of the method [150,155].

##### Release of Insects Carrying Dominant Lethality (RIDL)

In contrast to SIT, this method applies the use of a transgene which inhibits the larva and pupa stages from developing into the adult stage [156]. This lethal gene is repressible and can be regulated by an activator (tetracycline) before the engineered males are released to mate with the wild-type females. Field trials involving RIDL have been demonstrated in the Cayman Islands [157] and Brazil [158] using a modified strain *A. aegypti* OX513A with high suppression results of the wild-type *A. aegypti* [150].

Application of genetically modified mosquitoes is a new technology and not yet established in Africa. However, this does not rule out the fact that it could be the future of vector control and sub-Saharan countries need to move with the wave of advanced technology. It is, therefore, a challenge to the government to invest in such studies/programs in the effort towards control of mosquito vectors.

## 5. Conclusions

From the analyzed data, it is quite evident that the mosquito burden in Kenya cannot be underrated, especially with the multiple primary and secondary vectors roaming the fields and houses as well as the recurrent mosquito-borne disease outbreaks. Mosquito surveillance in the country has detected some species in regions which had no previous records probably due to the increased global warming and climate change. As a consequence, the actual mosquito burden in the country is not clearly understood. Many studies have focused mainly on zones of disease outbreaks overlooking other regions in which mosquito pathogens may be in silent sylvatic circulation waiting to explode in human populations. Regardless of the multiple resources allocated to the vector control programs which has seen a considerable vector reduction, more needs to be done to achieve actual eradication success. This includes integrating new ecologically benign and low resistant bio-controls such as entomopathogens into the large-scale vector management programs. In addition, it includes funding work focusing on the genetic engineering of mosquitos like SIT and RIDL. These measures would not only provide a sustainable solution to the mosquito menace but also reduce the use of synthetic pesticides which have been linked to significant environmental pollution and observed mosquito resistance.

## Figures and Tables

**Figure 1 insects-10-00173-f001:**
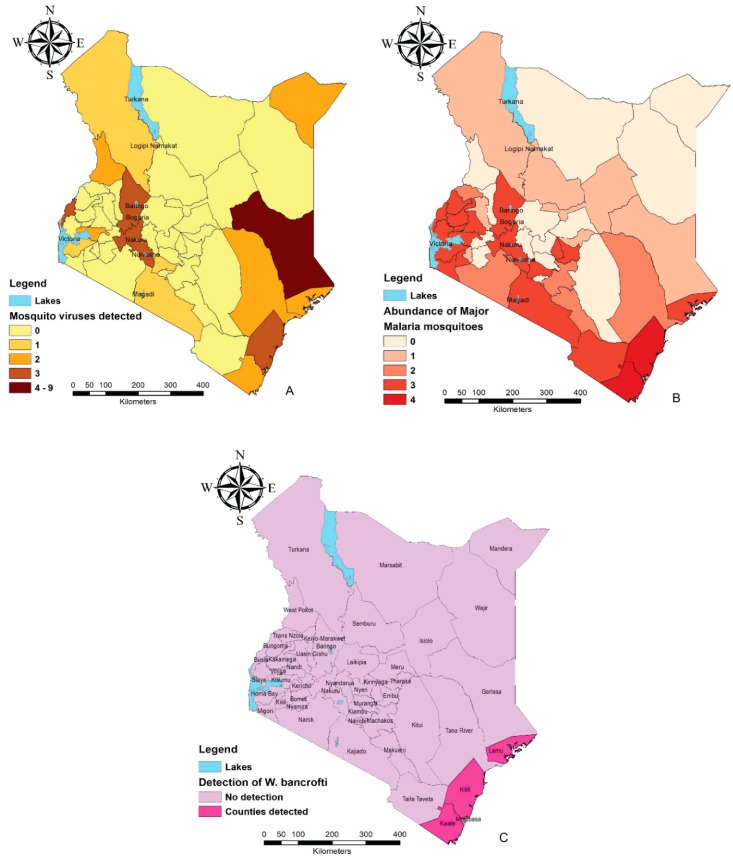
Mosquito-borne disease endemic regions based on distribution of pathogens and associated mosquito species (maps were constructed using the free and open-source Quantum GIS software (https://qgis.org/en/site/) using data compiled from Table 1 and Table 2. (**A**) Abundance of mosquito-borne viruses detected/isolated in various counties. (**B**) Distribution of the major malaria vectors in numbers in different counties. (**C**) Counties in which *Wuchereria bancrofti* has been detected from mosquitoes.

**Table 1 insects-10-00173-t001:** Summary of the main mosquito species in which viruses have been detected/isolated in Kenya.

Genera	Species	Virus Isolated/Detected ^1^	County of Virus Detection	Reference
*Aedes*	*A. aegypti*	DENV, CHKV	Mombasa, Mandera, Kilifi, Lamu, Busia	[12,13,33,55,57,58]
	*A. africanus*	YFV	Baringo,	[59]
	*A. albicosta*	DENV, CHKV	Mombasa, Kilifi, Lamu, Kwale	[33]
	*A. circumluteolus*	RVFV, BBKV, NDUV, SMFV	Garissa	[4,36]
	*A. fryeri*	DENV	Mombasa, Kilifi, Lamu, Kwale	[33]
	*A. fulgens*	DENV, CHKV	Mombasa, Kilifi, Lamu, Kwale	[33]
	*A. keniensis*	YFV	Baringo	[59]
	*A. Luridus*	NDUV	Tana River	[4]
	*A. mcintoshi*	RVFV, NDUV, PGAV, BUNV, BBKV, PGAV, SMFV, NRIV	Garissa	[4,11,36,40]
		DENV, CHKV	Mombasa, Kilifi, Lamu, Kwale	[33]
	*A. ochraceus*	RVFV, NDUV, BUNV, BBKV, SNBV, SMFV	Garissa	[4,11,36,37,38]
		DENV, CHKV	Mombasa, Kilifi, Lamu, Kwale	[33]
	*A. pembaensis*	RVFV	Kilifi	[4]
		DENV, CHKV	Mombasa, Kilifi, Lamu, Kwale	[33]
	*A. sudanensis*	BBKV, SNBV, WNV	Garissa	[36]
		NDUV	Tana River	[39]
*Anopheles*	*An. funestus*	ONNV	Kisumu	[59]
		BUNV	Kajiado	[4]
		NRIV	Tana River	[4]
	*An. gambiae*	BUNV	Homabay	[20]
	*An. squamosus*	RVFV	Garissa	[11]
*Culex*	*Cx. bitaeniorhynchus*	RVFV	Kilifi	[11]
		NDUV	Tana River	[39]
	*Cx. cinereus*	NDUV	Busia	[4]
	*Cx. pipiens*	USUV	Kisumu	[4]
		NDUV	Garissa, Tana River	[38,39]
	*Cx. poicilipes*	RVFV	Kilifi	[11]
	*Cx. quinquefasciatus*	RVFV	Baringo, Garissa	[11,60]
		WNV, SNBV	Garissa	[36,60]
	*Cx. rubinotus*	NDUV	Baringo	[4]
	*Cx. univittatus*	RVFV	Baringo	[11]
		BUNV	Homa Bay	[20]
		SNBV	West Pokot, Nakuru, Busia	[4,37,61]
		WNV	Garissa, Turkana, West Pokot	[4,61]
	*Cx. vansomereni*	NDUV	Tana River	[39]
		BBKV, SNBV	Nakuru	[4,37]
	*Cx. zombaensis*	RVFV	Nakuru	[62]
		BBKV	Kiambu	[4]
*Mansonia*	*Mn. africana*	RVFV	Nakuru, Baringo, Garissa	[11,60,62]
		NDUV	Baringo	[4]
	*Mn. uniformis*	RVFV	Baringo, Garissa	[40,60]
		NDUV	Baringo	[36]

Genera: *A, Aedes; An, Anopheles; Cx, Culex; Mn, Mansonia.*
^1^ Abbreviations: DENV—dengue virus, CHKV—chikungunya virus, RVFV—Rift Valley fever virus, NDUV—Ndumu virus, PGAV—Pongola virus, BUNV—Bunyamwera virus, BBKV—Babanki virus, SMFV—Semliki Forest virus, NRIV—Ngari virus, YFV—yellow fever virus, SNBV—Sindbis virus, WNV—West Nile virus, USUV—Usutu virus, ONNV—O’nyong’nyong virus.

**Table 2 insects-10-00173-t002:** Summary of the dominant malaria and Bancroftian filariasis mosquito vectors distributed in Kenya.

Parasite	Associated Human Ailment	Dominant Mosquito Species	Counties of Vector Distribution	Reference
***Plasmodium falciparum***	Malaria	*An. gambiae s.s., An. arabiensis, An. funestus, An. merus*	Kwale, Kilifi	[71,75,81,93,94]
		*An. gambiae ss, An. arabiensis, An. funestus*	Taita-Taveta, Lamu, Kajiado, Embu, Nakuru, Baringo, Bungoma, Kirinyaga, Kiambu, Busia, Siaya, Kakamega, Vihiga, Homabay, Migori, Kisii, Kisumu, Nandi	
		*An. gambiae ss, An. arabiensis*	Narok	
		*An. arabiensis, An. funestus*	Tana-River, Makueni, Machakos, Trans-Nzoia	
		*An. funestus*	Samburu, Isiolo, Garissa, Mombasa, Uasin-Gishu, Nyamira	
		*An. arabiensis*	Turkana	
		*An. gambiae ss*	Tharaka-Nithi	
***Wuchereria bancrofti***	Bancroftian filariasis	*An. gambiae sl, An. funestus, Cx. quinquefasciatus*	Kwale, Kilifi, Lamu	[50,91,95]

Genera: *An*, *Anopheles*; *Cx*, *Culex*.

**Table 3 insects-10-00173-t003:** Summary of the common mosquito control strategies in Kenya and proposed alternatives.

**Common Current Strategies**					
**Strategy**	**Description/products**	**Life stage target ^1^**	**Development of resistance**	**Advantage**	**Major challenges**
**Surveillance**	Use mosquito collection tools; (CO_2_-baited CDC (Centers for Disease Control) light traps and 350 mL larval dippers)	L, P, A	None	Identify mosquito distribution status (guide on control measures)	Time consuming
**Synthetic pesticides**	Involve chemicals (pyrethroids, carbamates ^B^, organophosphates ^B^, and organochlorines ^B^) integrated in IRS and LLIN	L, P, A	High	Fast mode of action, broad spectrum	Environmental unfriendly, toxic to non-target biota and easy to develop resistance
**Insect growth regulators (IGRs)**	Involve chemicals or plant extracts to inhibit metamorphosis. (methoprene and pyriproxyfen)	E, L, P, A	Low	Safe to non-target organisms, target specific	Slow mode of action
**Alternative Strategies**					
**Public awareness**	Involving the public in clearing nearby breeding zones (open water containers, tires, tins, bottles, etc.)	E, L, P, A	None	Broad spectrum, minimal use of pesticides	Lack of funding for mass education countrywide
**Biological control**	Natural predators that feed on mosquito (Larvivorous fish, Omnivorous copepods, amphibians, etc.)	E, L, P, A	None	Broad spectrum, environmentally friendly	Limited area of applicability, threat to few non-target organisms
	Entomopathogenic bacteria (*Bacillus thuringiensis vr israelensis* and *Lysinibacillus sphaericus*)	L	Low	Species specific, harmless to non-target organisms	Infective in cryptic breeding sites, limited to larvae control, active only after ingestion
	Entomopathogenic fungi (*Beauveria bassiana* and *Metarhizium anisopliae*)	L, A	Low	Broad spectrum, environmentally friendly	Cost ineffective, limited duration of efficacy
	Infecting male mosquito with *Wolbachia* (wmel *Wolbachia* strain)	A	Low	Species specific, environmentally friendly	Mating competitiveness, high cost of male mosquito rearing
**Genetic engineering**	Sterile insect technique (SIT)—Genetic suppression of the male mosquito	A	None	Species specific, environmentally friendly	Mating competitiveness, high cost of male mosquito rearing and molecular tools
	Release of insects carrying dominant lethality (RIDL)—transgenic prevention of adult development	A	None	Species specific, environmentally friendly	Mating competitiveness, high cost of male mosquito rearing and molecular tools

Abbreviations: E, egg; L, larvae; P, pupa; A, adult; IRS, indoor residual spraying; LLIN, long-lasting insecticide nets. ^B^ Banned in Kenya.

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
