# Peer review of "Mosquitoes of Etiological Concern in Kenya and Possible Control Strategies"

_insects, 2019, doi:10.3390/insects10060173_

Round 1

Reviewer 1 Report

I have reviewed the manuscript titled “Mosquitoes of Etiological Concern in Kenya and Possible Control Strategies” by Karungu et al. The paper describes the current understandings of mosquito health risks in Kenya and the strategies available for addressing those risks.

Papers of this nature are useful in providing documentation of local mosquito issues and can be used as leverage to assist local authorities justify strategies for the surveillance and control of local mosquito populations and initiate programs that may otherwise assist in reducing the burden of mosquito-borne disease. The framework for the paper is strong with the summaries provided in Tables 1, 2 and 3 providing an excellent foundation but the presentation of the manuscript is deficient in its current state.

The sections detailing current control strategies can be greatly improved. This is especially the case where there are distinct approaches used to address mosquito population control associated with the arboviruses and malaria parasites. A clearer distinction can be made here. I also note that there is no reference (that I could find) to the use of insect growth regulators in mosquito control, these are commonly used across the world to control a wide range of mosquitoes of pest and public health concern.

Overall, I feel that the manuscript could benefit from review with regard to presentation. There are many occasions where the writing style is not consistent with scientific literature. This makes a thorough review difficult at this time and substantial revision of manuscript is required.

There is also the lack of inclusion of authorities for individual species and scientific names are not italicised. Conversely, there are occasions where italics are used incorrectly (e.g. “species” or “spp.”). I feel it is beyond the scope as a reviewer to work through the whole manuscript and offer these detailed changes due to sheer volume of revisions required but encourage the authors to seek out an individual who may be able to assist.

Author Response

Point 1: The sections detailing current control strategies can be greatly improved. This is especially the case where there are distinct approaches used to address mosquito population control associated with the arboviruses and malaria parasites. A clearer distinction can be made here.

Response 1: Various approaches have previously been discussed in depth covering the scope of mosquito control which several references have been cited in our current review. Our main focus was to evaluate the situation in Kenya in terms of mosquitos of public health concern and a summary of both current and alternative control measures. In most parts of the mosquito control section, the strategies have been highlighted and target mosquito species mentioned.

To additionally reinforce this section, target mosquito species have also been added in the sub-sections “4.2.2.2. Largescale applications of entomopathogens and 4.2.3.1. The Sterile Insect Technique (SIT)” highlighted in green.

Point 2: I also note that there is no reference (that I could find) to the use of insect growth regulators in mosquito control, these are commonly used across the world to control a wide range of mosquitoes of pest and public health concern.

Response 2: This has been added as subsection “4.1.3. Insect growth regulators (IGRs) as well as in Table 3”.

Point 3: Overall, I feel that the manuscript could benefit from review with regard to presentation. There are many occasions where the writing style is not consistent with scientific literature. This makes a thorough review difficult at this time and substantial revision of manuscript is required.

Response 3: The whole manuscript has been revised to address the concern highlighted above. The changes have been set in tracking mode for easy follow up.

Point 4: There is also the lack of inclusion of authorities for individual species and scientific names are not italicised. Conversely, there are occasions where italics are used incorrectly (e.g. “species” or “spp.”). I feel it is beyond the scope as a reviewer to work through the whole manuscript and offer these detailed changes due to sheer volume of revisions required but encourage the authors to seek out an individual who may be able to assist.

Response 4: The issue of inclusion of authorities was quite a challenge bearing in mind that we based our review on published articles. As a result, we stated the species names as they were mentioned in cited articles.

As for italics being used incorrectly, this has been corrected and scientific names italicized properly which can be verified via tracked changes. 

Reviewer 2 Report

Overall, this review article gave a comprehensive description of mosquito and it's related diseases in Kenya, The vector control approaches have been mentioned in the article as well. 

This article is worth to publish in the journal if some statement has been modified.

Lines 378-381 In fact, DDT is still allowed to be used for IRS in Africa countries to combat malaria. Please modify this statement. 

Please make sure that all the species names are in the italic form.

Author Response

Response to Reviewer 2 Comments

 Point 1: Lines 378-381 In fact, DDT is still allowed to be used for IRS in Africa countries to combat malaria. Please modify this statement. 

Response 1: This statement was made in reference to Kenya and not Africa in general. The following statement highlighted in blue in the manuscript has however been Modified to clarify that only pyrethroids are allowed as pesticides for use in Indoor residual spraying (IRS) and long-lasting insecticide nets (LLIN) in Kenya. A citation has also been provided. 

Point 2: Please make sure that all the species names are in the italic form.

Response 1: This has been corrected as per your suggestion. Changes have been tracked accordingly in the Manus.

Thank you

Reviewer 3 Report

General assessment and comments:

In the submitted manuscript, Karungu et al. reviewed current knowledge on mosquitoes as infectious diseases vector and mosquito control strategies in Kenya. The authors first discussed the prevalence and geographical distribution of mosquito-borne diseases in Kenya. In addition, authors summarized the mosquito genera in Kenya and viruses that are isolated from each genera. Furthermore, the authors discussed the current control strategies and potential measures that could be implanted in the future. 

This review will provide helpful information to the readers in the field who are interested in this topic. Overall, this review is comprehensive and well written.  

Author Response

Response to reviewer 3  comments.

Point 1: This review will provide helpful information to the readers in the field who are interested in this topic. Overall, this review is comprehensive and well written.  

Response 1: Thank you for the comment.

Round 2

Reviewer 1 Report

I thank the authors for addressing my comments on the originally submitted manuscript. The presentation of the revised manuscript is much improved.

The only minor comments to make are with regard to the placement of tables in text. The placement should be modified to ensure that the caption for Table 2 is on the same page as the table itself and, if possible, could Table 3 fit on a single page without needing to be split? I would suggest modification of column widths to achieve this (especially column 2) 

Author Response

Point by point response to Reviewer 1

Comment 1

The placement should be modified to ensure that the caption for Table 2 is on the same page as the table itself.

Response 1

This has been corrected as per your suggestion in the manuscript. Table 2 Highlighted in purple.

Comment 2

if possible, could Table 3 fit on a single page without needing to be split? I would suggest modification of column widths to achieve this (especially column 2).

Response 2

This has been corrected as per your suggestion in the manuscript. Table 3 Highlighted in purple.
